# ATTENTION FLOWS FOR GENERAL TRANSFORMERS

## ABSTRACT

In this paper, we study the computation of how much an input token in a Transformer model influences its prediction. We formalize a method to construct a flow network out of the attention values of encoder-only Transformer models and extend it to general Transformer architectures, including an auto-regressive decoder. We show that running a maxflow algorithm on the flow network construction yields Shapley values, which determine a player's impact in cooperative game theory. By interpreting the input tokens in the flow network as players, we can compute their influence on the total attention flow leading to the decoder's decision. Additionally, we provide a library that computes and visualizes the attention flow of arbitrary Transformer models. We show the usefulness of our implementation on various models trained on natural language processing and reasoning tasks.

## 1 INTRODUCTION

The *Transformer* (Vaswani et al., 2017) is the dominant machine learning architecture in recent years, finding application in NLP (e.g., BERT (Devlin et al., 2019), GPT-3 (Brown et al., 2020), or LaMDA (Collins and Ghahramani, 2021)), computer vision (see Khan et al. (2021) for a survey), mathematical reasoning (Lample and Charton, 2019; Han et al., 2021), or even code and hardware synthesis (Chen et al., 2021; Schmitt et al., 2021). The Transformer relies on *attention* (Bahdanau et al., 2015) that mimics cognitive attention, which sets the focus of computation on a few concepts at a time. In this paper, we rigorously formalize constructing a flow network out of attention values (Abnar and Zuidema, 2020) and generalize it to models, including a decoder. While theoretically yielding a Shapley value (Shapley, 1953) quite trivially, we show that this results in meaningful explanations for input tokens' influence on the total flow affecting a Transformer's prediction.

Its applicability in various domains has made the Transformer architecture incredibly popular. Models are easily accessible for developers around the world, for example at `huggingface.co` (Wolf et al., 2019). However, blindly using or fine-tuning these models might lead to mispredictions and unwanted biases, which will have a considerable negative effect on their application domains. The sheer size of the Transformer models makes it impossible to analyze the networks by hand. Explainability and visualization methods, e.g., Vig (2019), aid the machine learning practitioner and researcher in finding the cause of a misprediction or revealing unwanted biases. The training method or the dataset can then be adjusted accordingly.

Abnar and Zuidema (2020) introduced *Attention Flow* as a post-processing interpretability technique that treats the self-attention weight matrices of a Transformer encoder as a flow network. This technique allows analyzing the flow of attention through the Transformer encoder: Computing the maxflow for an input token determines the impact of this token on the total attention flow. Ethayarajh and Jurafsky (2021) discussed a possible relation of the maxflow computation through the encoder flow network to Shapley values, which is a concept determining a player's impact in cooperative game theory and can be applied to measure the importance of a model's input features. However, the lack of a clear formalization of the underlying flow network has made it difficult to assess the validity of their claims, which we aim to address in this work.

We extend our formalization of the approach to a Transformer-model-agnostic technique, including general encoder-decoder Transformers and decoder-only Transformers such as GPT models Radford et al. (2018). While, after applying a positional encoding, the encoder processes the input tokens as a whole, the decoder layers operate auto-regressively, i.e., a sequence of tokens will be predicted step-by-step, and already predicted input tokens will be given as input to the decoder. This results in a

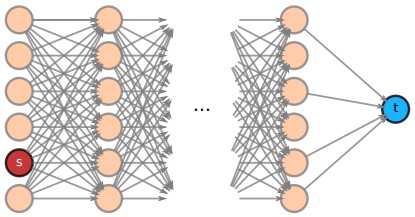

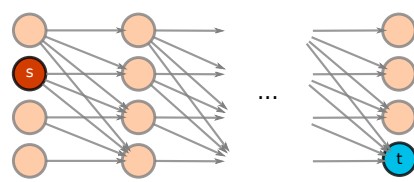

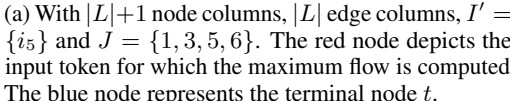

(a) With $|L|+1$ node columns, $|L|$ edge columns, $I' = \{i_5\}$ and $J = \{1, 3, 5, 6\}$. The red node depicts the input token for which the maximum flow is computed. The blue node represents the terminal node $t$.

(b) Input token set $O' = \{o_2\}$ and embedding $t$, where the output token $o_5$ is currently predicted. The first "input" token, i.e., $o_1$ is the special start token of the decoder.

Figure 1: Encoder attention in a flow network (left) and decoder attention in a flow network (right).

significantly different shape of the flow network and, in particular, requires normalization to account for the bias towards tokens that were predicted later than others. We account for the auto-regressive nature of the decoder by ensuring *positional independence* of the computed maxflow values. We implemented our constructions as a Python library, which we will publish under the MIT license. In summary, our contributions are the following. We formalize encoder-only attention flow and generalize the approach to encoder-decoder and decoder-only Transformers in Section 2. Furthermore, we use the formalization to construct an explicit algorithm for attention flow computation and analyze its complexity. In Section 3, we show that the computed attention flow values are Shapley values for all three architectures. Section 4 introduces a tool to compute and visualize attention flow for arbitrary Transformers. We report on qualitative and quantitative experiments that show the effectiveness of our approach, including token bias and single-head attention analyses.

*Related Work.* We would like to emphasize the work on which we build: Abnar and Zuidema (2020) introducing attention flow for Transformer encoders and Ethayarajh and Jurafsky (2021) drawing a possible connection between encoder attention flows and Shapley values. An explainability overview is given by Samek et al. (2017) and Burkart and Huber (2021). An overview over Shapley value formulations for machine learning models is given by Sundararajan and Najmi (2020), which are not restricted to Transformer models and do not include attention flow (Lindeman, 1980; Grömping, 2007; Owen, 2014; Owen and Prieur, 2017; Štrumbelj et al., 2009; Štrumbelj and Kononenko, 2014; Datta et al., 2016; Lundberg and Lee, 2017; Lundberg et al., 2018; Aas et al., 2019; Sun and Sundararajan, 2011; Sundararajan et al., 2017; Agarwal et al., 2019). Shapley values are also used for the valuation of machine learning data (Ghorbani and Zou, 2019). Raw attention values can be visualized, e.g., Vig (2019) and Wang et al. (2021). Chefer et al. (2021) assign local relevance based on the Deep Taylor Decomposition principle (Montavon et al., 2017).

## 2 ATTENTION FLOW

*Attention Flow* (Abnar and Zuidema, 2020) is a post-processing interpretability technique that treats the self-attention weight matrices of the Transformer encoder as a flow network and returns the maximum flow through each input token. Formally, a flow network is defined as follows.

**Definition 1** (Flow Network). *Given a graph $G = (V, E)$, where $V$ is a set of vertices and $E \subseteq V \times V$ is a set of edges, a flow network is a tuple $(G, c, s, t)$, where $c : E \to \mathbb{R}_\infty$ is the capacity function and $s$ and $t$ are the source and terminal (sink) nodes respectively. A flow is a function $f : E \to \mathbb{R}$ satisfying the following two conditions: Flow conservation: $\forall v \in V \setminus \{s, t\}. \ x_f(v) = 0$, where $x_f : V \to \mathbb{R}$ is defined as $x_f(u) = \Sigma_{v \in V} f(v, u)$, and capacity constraint: $\forall e \in E. \ f(e) \leq c(e)$.*

The value of flow $|f|$ is the amount of flow from the source node $s$ to the terminal node $t$: $|f| = \sum_{v:(s,v) \in E} f_{sv}$. For a given set $K$ of nodes, we define $|f(K)|$ as the flow value from $s$ to $t$ only passing through nodes in $K$: $|f(K)| = \sum_{v:(s,v) \in E, v \in K} f_{sv}$. We define $|f_o(v)|$ to be the total outflow value of a node $v$ and $|f_i(v)|$ to be the total inflow value of a node $v$. In optimization theory, the maximum flow problem $max(|f|)$ (Harris and Ross, 1955) is to find flows that push the maximum possible flow value $|f|$ from the source node $s$ to the terminal node $t$, which we denote by $f_{max}$.

## 2.1 ENCODER ATTENTION FLOW

Given an encoder-only Transformer model, such as the BERT (Devlin et al., 2019) model family, with $H$ attention heads, $L$ layers, $M$ input tokens $I = \{i_1, \ldots, i_M\}$ and the resulting self-attention tensor $\mathbf{A}^E \in \mathbb{R}^{H \times L \times M \times M}$. For some $X \in \mathbb{N}$, we define $[X]$ as the set $\{1, \ldots, X\}$. For a set of positions $J$, a subset of input tokens $I' \subseteq I$ and subset of heads $H' \subseteq H$, we construct a flow network $\mathcal{F}_{enc}(\mathbf{A}^E, I', J) = (G, c, s, t)$ as follows:

$$V := (I \times [L+1]) \cup \{s, t\} \ , \qquad c((i_j, l), v') := \begin{cases} \frac{1}{H'} \sum_{h=1}^{H'} \mathbf{A}^E_{h,l,k,j} & v' = (i_k, l+1) \\ \infty & v' = t \end{cases} \ ,$$

$$E := \{((i_j, l), (i_k, l+1)) \mid i_j, i_k \in I \wedge l \in [L+1]\} \cup \{((i_j, L+1), t) \mid i_j \in I \wedge j \in J\}$$
$$\cup \{(s, (i', 0)) \mid i' \in I'\} \ .$$

We visualize this flow network translation in Figure 1a. The flow network consists of $L+1$ columns of nodes and $L$ columns of edges. The attention values are encoded as capacities on the edges. Thus the underlying graph of the flow network requires one additional column of nodes. Computing the maximum flow through this network determines the contribution of the input tokens $I'$ to the attention flow towards the final encoder embeddings given by $J$. Note that the nodes in columns greater than 1 correspond to encoder embeddings and can not be interpreted as input tokens anymore. Residual connections can be taken into account as proposed by Abnar and Zuidema (2020), i.e., by adding an identity matrix $I$ and re-normalize it as $0.5\mathbf{A} + 0.5I$. By setting the start node $s$ successively to singleton sets containing only a single input token and all final embeddings to $t$, we can compute the encoder flow for every encoder input token as introduced by Abnar and Zuidema (2020). The encoder flow network construction can also be used for models including a classification task (see Section 4). To determine the influence of input tokens on the attention flow towards deciding the class, the terminal node $t$ is only connected to the final embedding of the classification token.

## 2.2 DECODER ATTENTION FLOW

Generative Transformer models that involve a decoder require a significantly different shape of flow network. We begin by investigating decoder-only models, with $H$ attention heads, $L$ layers, $N$ "output" tokens $O = \{o_1, \ldots, o_N\}$ and the self-attention tensor $\mathbf{A}^D \in \mathbb{R}^{H \times L \times N \times N}$. Since we consider decoder-only models, a prefix subset $O_{input} \subseteq O$ will be given as a problem input to the neural network model. Note that the first output token is always a special start token. For a set of output tokens $O' \subseteq O$, the position $n$ of output token $o_n \in O$ and subset of heads $H' \subseteq H$, the construction of a flow network $\mathcal{F}_{dec}(\mathbf{A}^D, O', n) = (G, c, s, t)$ follows the structure of the decoder self-attention:

$$V := O \times [L+1] \ , \qquad E := \{(o_j, l), (o_k, l+1)) \mid o_j, o_k \in O \wedge l \in [L+1] \wedge j \leq k\} \ ,$$

$$c((o_j, l), (o_k, l+1)) := \frac{1}{H'} \sum_{h=1}^{H'} \mathbf{A}^D_{h,l,k,j} \ , \qquad s := \{(s, (o', 0)) \mid o' \in O'\} \ , \qquad t := (o_{n-1}, L+1) \ .$$

We visualize the construction in Figure 1b. Because of the auto-regressive nature of the Transformer decoder, we compute the maxflow to the last embedding of the decoder as this embedding will be used in the Transformer to predict the next token. The auto-regression, however, requires a normalization to account for the bias towards tokens that were predicted later than others (later predicted tokens have more incoming edges). Intuitively, we require that the maxflow computation for any sub flow network $F'$ constructed from the decoder flow network $F$ to be independent of the absolute position of $F'$ in $F$. Formally, assuming $\mathbf{A}^D$ to have the same value $c$ for every entry, i.e., the capacity of every edge in the resulting-flow network is fixed to $c$, we require for every position $n$ that $\forall o_m \in O.\ maxflow(\mathcal{F}_{dec}(\mathbf{A}^D, \{o_m\}, n)) = c$, which we call *positional independence*. We ensure this by dividing the result of a max flow computation for a given start token $o_m$ and end token $o_n$ by $1 + (O - (n - m)) - m$. For a subset $O' \subseteq O$ and a position $n$ (where $\forall o'_m \in O'.m < n$) and heads $H'$, we can thus compute the influence of the token set $O'$ to the total attention flow towards the embedding that predicts the $n$-th token, no matter whether it served as part of the problem input or is an already predicted output token.

## 2.3 ENCODER-DECODER ATTENTION FLOW

For Transformer models consisting of an encoder and a decoder, we combine both flow network translations with the encoder-decoder attention. Figure 2 shows the structure of the flow network for a Transformer model with an encoder (top) and a decoder (bottom). The last nodes of the flow network corresponding to the final embedding of the encoder are, following the Transformer architecture, connected to every node layer of the network corresponding to the decoder. We omit some encoder-decoder edges for better visualization. Given a Transformer with $H$ attention heads, $L$ layers, $M$ input tokens $I = \{i_0, \ldots, i_M\}$, $N$ output tokens $O = \{o_0, \ldots, o_N\}$, and resulting encoder self-attention tensor $\mathbf{A}^E \in \mathbb{R}^{H \times L \times M \times M}$, decoder self-attention tensor $\mathbf{A}^D \in \mathbb{R}^{H \times L \times N \times N}$ and encoder-decoder attention tensor $\mathbf{A}^C \in \mathbb{R}^{H \times L \times N \times M}$. For a set of input tokens $I'$, the position $n$ of output token $o_n$ and subset of heads $H' \subseteq H$, we construct a flow network $\mathcal{F}(\mathbf{A}^E, \mathbf{A}^D, \mathbf{A}^C, I', n) = (G, c, s, t)$ from flow networks $\mathcal{F}_{enc}(\mathbf{A}^E, I', \emptyset) = ((V_{enc}, E_{enc}), c_{enc}, s_{enc}, t_{enc})$ and $\mathcal{F}_{dec}(\mathbf{A}^D, \emptyset, n) = ((V_{dec}, E_{dec}), c_{dec}, s_{dec}, t_{dec})$ as follows:

$$V := V_{enc} \cup V_{dec} \cup s \ , \qquad E := E_{enc} \cup E_{dec} \cup \{((i_j, L+1), v) \mid i_j \in I \wedge v \in V_{dec}\}$$
$$t := (o_n, L+1) \ , \qquad\qquad \cup \{(s_{enc}, (o_m, 0)) \mid o_m \in I \wedge m < n\} \ ,$$

$$c(v, v') := \begin{cases} c_{enc}(v, v') & v = (i_j, l), v' = (i_k, l'), \\ & \quad i_j, i_k \in I \\ c_{dec}(v, v') & v = (o_j, l), v' = (o_k, l'), \\ & \quad o_j, o_k \in O \\ \frac{1}{H'} \sum_{h=1}^{H'} \mathbf{A}^C_{h,l,k,j} & v = (i_j, L+1), \\ & \quad v' = (o_k, l), i_j \in I, o_k \in O \\ \infty & v = s, v' \in I' \end{cases} \ ,$$

where $l_e$ denotes a layer from the encoder and $v_d$ denotes a node from the decoder. Again, we have to normalize to account for the auto-regressive bias, i.e., require positional independence. For a given set on input tokens $I'$ and heads $H'$, we can thus asses the contribution of this set to the total attention flow towards the embedding that predicts the $n$-th token by computing the maxflow through this network. If one is interested in the influence of an already computed output token $o_m$, where $m < n$, on the prediction of $o_n$, then the construction for the decoder-only case in Section 2.2 applies.

## 2.4 ALGORITHM

The flow network constructions can be directly used in an algorithm to compute the attention flow for input tokens. Algorithm 1 shows the algorithm for computing the attention flow for every input-output token pair. We build the flow network for every pair and compute the maximum flow in the corresponding network with the Edmonds-Karp Edmonds and Karp (1972) algorithm. The runtime of Edmonds-Karp is in $O(VE^2)$, where the edges $E$ and nodes $V$ are given by the number of layers and input-output tokens. Since we run this algorithm for every input-output

**Input:** $\mathbf{A}^E, \mathbf{A}^D, \mathbf{A}^C, I, O$
**Output:** $f : O \times I \rightarrow \mathbb{R}$
$f = \text{None}$
**for** $o \in O$ **do**
    **for** $i \in I$ **do**
        $f(o, i) \leftarrow$
        $Ed.Ka.(\mathcal{F}(\mathbf{A}^E, \mathbf{A}^D, \mathbf{A}^C, \{i\}, o))$
**return** f

**Algorithm 1:** Attention flow.

pair (with only partially rebuilding the flow network), we additionally gain linear complexity in the number of input and output tokens. We evaluate the implementation of this algorithm and its variations for encoder-only and decoder-only Transformers in Sec. 4.

## 2.5 OPTIMIZATIONS

The flow network constructions apply to subsets of heads, especially single heads. The results of head computations are joined using a linear projection, so each head has access to the computations of all heads in the previous layers. The task of a head in layer $l$ can be independent of its task in previous layers $l' < l$. In practice, however, heads are biased towards keeping their respective tasks, such that we also found good interpretability results by considering the attention flow of attention heads independently (see Section 4). A flow network can be constructed for a single head by following the above constructions, setting $H'$ to every singleton. If the computation time of the maxflow for large

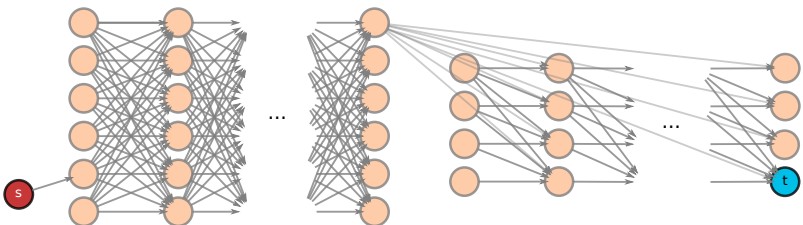

Figure 2: Sketch of the Encoder-decoder attention flow network for input token $i_5$ and embedding $t$, which is used to predict $o_5$. Encoder-decoder connections are sketched for the first node.

Transformer models exceeds time limits, relaxations on the flow network are possible. First, note that the flow network only needs to be constructed once. As expected, the computation time of the maxflow in the network constructions increases with larger input and output sequences. Running time can be traded against heuristically shrinking the size of the flow network. This can be done in two dimensions. Following the practical assumption that heads often keep their tasks throughout subsequent layers, the first dimension is to shrink the flow network on the $x$-axis. This can be done by simply skipping some of the inner layers of the network or by merging layers by taking the average of the raw attention values across layers as capacities. Furthermore, the network can also be shrunk in the $y$-dimension similarly by grouping input and output tokens. For example, tokens *predict* and *ed* can be combined into one node.

## 3 SHAPLEY VALUE EXPLANATIONS

In this section, we show how the extended flow network constructions over the Transformer decoder $\mathcal{F}_{dec}(\mathbf{A}^D, O', n)$ and $\mathcal{F}(\mathbf{A}^E, \mathbf{A}^D, \mathbf{A}^C, I', n)$ induce Shapley value explanations for the tokens of the input sequence. The Shapley value (Shapley, 1953) is a solution concept determining the impact of a player in cooperative game theory and an increasingly popular concept to determine the influence of certain input features on a model's decision.

**Definition 2.** *A game with transferable utility (TU) is a pair $(P, v)$, with $P = \{1, \ldots, p\}$ being a finite set of players and $v : 2^P \to \mathbb{R}$ being the payoff function.*

A subset $S \subseteq P$ is called a *coalition*. The payoff function $v$ assigns every coalition of players $S$ a real number $v(S) \in \mathbb{R}$ with $v(\emptyset) = 0$. The *share* of a player $i$ of the allocated payoff is $\varphi_i(v)$. The encoding of the attention values as a flow network is a TU game. A node in the flow network represents a player and the total flow through the network represents the total payoff (Ethayarajh and Jurafsky, 2021). The Shapley values of the players in a TU game are formally defined as follows.

**Definition 3** (Shapley Value). *Let $\Pi(P)$ be the set of all player permutations and let $\pi \in \Pi(P)$ be a permutation of players. Let all players ahead of a player $i$ be defined as $P_{<i}(\pi) := \{j \in P : \pi(j) < \pi(i)\}$. The Shapley value $\varphi$ is defined as the share of payoff for a given player $i \in P$: $\varphi_i(P, v) := \frac{1}{p!} \sum_{\pi \in \Pi(P)} (v(P_{<i}(\pi) \cup \{i\}) - v(P_{<i}(\pi)))$.*

From a game-theoretic viewpoint, Shapley values are well-suited for determining the payoff share that players deserve, as the values satisfy the desirable properties efficiency, symmetry, null player, and additivity. The mathematical definition of the properties can be found in App. A These properties above are also responsible for making Shapley values an attractive approach for explaining a model's decisions, i.e., features that do not contribute to the accuracy of a model should be null players, and features that contribute equally should satisfy symmetry.

**Proposition 1** (Decoder-Only Flow Is a Shapley Value). *Consider a Transformer decoder with $H$ attention heads, $L$ layers, $N$ "output" tokens $O = \{o_1, \ldots, o_N\}$ and the self-attention tensor $\mathbf{A}^D \in \mathbb{R}^{H \times L \times N \times N}$. Let $f^o_{max}$ be the maxflow computed in the flow network $\mathcal{F}_{dec}(\mathbf{A}^D, \{o\}, n)$ as defined in the previous section. Consider the TU-game $(P, v)$, where the players $p \in P = \{1, ..., N\}$ correspond to nodes $(o_p, 0)$ from the first layer of the Transformer decoder. For a given coalition $S \subseteq P$, let the value function be $v(S) = \sum_{s \in S} f^{o_s}_{max}$, i.e., the sum of max-flows of nodes corresponding to $S$. Then, the max-flow $f^{o_s}_{max}$ for some $p \in P$ is its Shapley value.*

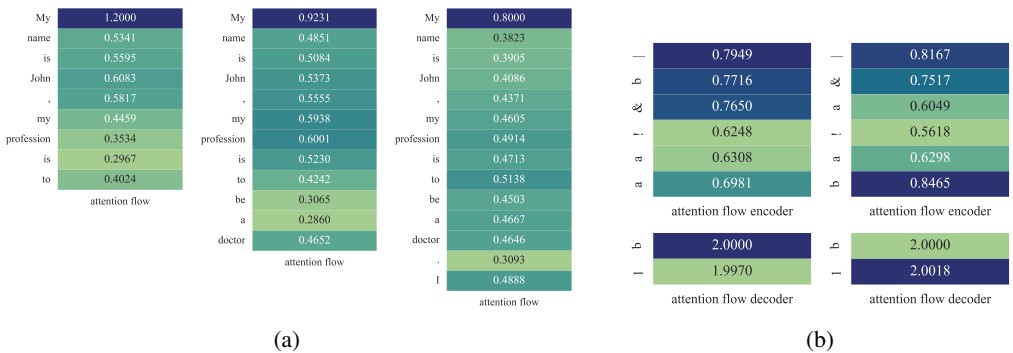

(a)                                                                          (b)

Figure 3: Heatmap of the attention flow of the GPT-2 model after $1$, $4$, and $6$ predicted tokens in (a) and Heatmap depicting the attention flow for unimportant token detection in SAT assignments in (b).

The proof immediately follows from the fact that every max-flow of some node $f^o_{max}$ is an independent computation and the payoff of a coalition is defined as a sum of these independent contributions, which trivially qualifies as a Shapley value. Although this theoretical correspondence to a Shapley value is trivial, we show in our experiments in the following section that the maxflow computation indeed yields meaningful explanations for the network's attention flow.

Note that our line of reasoning significantly differs from Ethayarajh and Jurafsky (2021). In particular, we compute a separate max-flow for every token in the set of players. This is because key assumptions about flow networks that they make in their proof do not hold: They argue that as long as nodes come from the same layer, blocking flow through some of these nodes does not change the possible flow through the others, such that they can deduce that the utility a player adds when joining a coalition is independent of the identity of the players already in the coalition. However, this is not the case: Several nodes from the same layer can compete for capacity downstream in the network even if they have no direct connection, e.g., if we have two tokens $o_1, o_2$ in one layer each attended to with $0.5$ attention by a node $o_3$ which itself is only attended to with $0.5$ attention. Now, the utility $o_1$ adds upon joining a coalition as defined by Ethayarajh and Jurafsky (2021) does depend on whether $o_2$ is already part of it. We deduce from the above discussion that it may violate the symmetry of a Shapley value, as the payoff for $o_1$ and $o_2$ can be unequally allocated.

The ideas outlined for Proposition 1 also apply to the encoder-decoder attention flow. In the following, let $f^i_{max}$ be the maxflow computed in the flow network construction $\mathcal{F}(\mathbf{A}^E, \mathbf{A}^D, \mathbf{A}^C, \{i\}, n)$ over the Transformer with $H$ attention heads, $L$ layers, $M$ input tokens $I = \{i_0, \ldots, i_M\}$, $N$ output tokens $O = \{o_0, \ldots, o_N\}$, and resulting encoder self-attention tensor $\mathbf{A}^E \in \mathbb{R}^{H \times L \times M \times M}$, decoder self-attention tensor $\mathbf{A}^D \in \mathbb{R}^{H \times L \times N \times N}$ and encoder-decoder attention tensor $\mathbf{A}^C \in \mathbb{R}^{H \times L \times N \times M}$.

**Corollary 2** (Encoder-Decoder Flow Is a Shapley Value). *Consider the TU-game $(P, v)$, where the players $p \in P = \{1, ..., N\}$ correspond to nodes $(i_p, 0)$ from the first layer. Let the value function or a given coalition $S \subseteq P$ be defined as $v(S) = \sum_{s \in S} f^{i_s}_{max}$, i.e., the sum of max-flows of nodes corresponding to $S$. Then, the max-flow $f^{i_s}_{max}$ for some $p \in P$ is its Shapley Value.*

## 4 EXPERIMENTS

In this section, we report on natural language processing and logical reasoning experiments. We implemented the algorithm from Section 2.[1] The architectural details of the models are shown in Table 1b. We visualize the maxflow attention values in heatmaps, lineplots, and violinplots (see, for example, Figure 3b). The maxflow is computed with NETWORKX Hagberg et al. (2008), and the heatmaps comparing the attention flow from input/predicted token to current predicted token are visualized with SEABORN Waskom (2021). The heatmaps are either showing only the attention flow from input tokens if the model is encoder-only (*enc.*), are separated into different heatmaps for input tokens and auto-regressive tokens for encoder and decoder (*enc. + dec.*), or show one heatmap for all tokens if the architecture is decoder only (*dec.*). Higher values represent higher attention flow.

---

[1]The code and experiments will be published after the double-blind review phase.

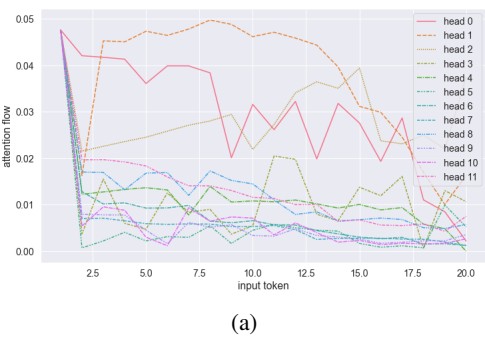
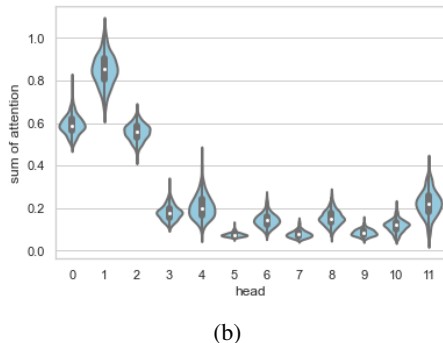

(a)                                             (b)

Figure 4: The attention flow for every head of GPT-2 separately in (a) and the sum of all attention flow values per head for 300 sampled input queries on GPT-2 in (b).

## 4.1 Token Relevancy in Decoder Flow

**Text completion.** We demonstrate that our technique captures the attention flow changes during auto-regressive decoding. For this experiment, we track the attention flow changes in GPT-2 Radford et al. (2019) while decoding the predicted tokens. The input sequence to this decoder-only mode was "My name is John, my profession is". Figure 3a depicts the attention flow after decoding the first, fourth, and sixth tokens. The resulting flow network can be found in Figure 13 in the appendix. Generally, GPT-2 models attend the first token the most (cf. Section 4.3). The differences in the attention flow are visible as the attention flow on previous tokens is different for each decoding step. Most notably, the attention flows shift heavily toward the token "profession" when predicting the token "doctor". We observed these heavy switches in decoder attention flow values throughout our experiments, which is why this approach is a valuable addition to existing analysis methods. The computation time of the flow values for this example only took 1.38, 1.50, and 2.09 seconds.

**Satisfying assignments for SAT.** In this experiment, we considered the problem of computing a satisfying assignment to a propositional logical formula. A formula in propositional logic is constructed out of variables and Boolean connectives $\neg$ (not), $\vee$ (or), $\wedge$ (and), $\rightarrow$ (implication), and $\leftrightarrow$ (equivalence). For example, let the following propositional formula be given: $b \vee (a \wedge \neg a)$. A satisfying assignment is a mapping from variables to truth values, such that the formula evaluates to true. For example, a satisfying assignment for the formula above is the mapping $\{b \mapsto 1, a \mapsto 0\}$. The variable $a$, however, has no impact on the truth value of the formula. As long as $b$ is set to $1$, $a$ can be predicted either as $1$ or $0$. We conducted an experiment to detect parts of the propositional formula that have no impact on predicted assignments. We trained a Transformer with an encoder and decoder to predict satisfying assignments. The attention flow values for the following two propositional formulas are depicted in Figure 3b: $PropSAT_1 := b \vee (a \wedge \neg a)$ in tokens: $b|(a\&!a)$ and $PropSAT_2 := (a \wedge \neg a) \vee b$ in tokens: $(a\&!a)|b$. The disjunct $(a \wedge \neg a)$ plays no role in any satisfying assignment since any mapping of $a$ results in this subformula being false. Regardless of the position in the formula, the flow computation of the network detects this as unimportant: the inputs to the encoder $a$ and $\neg a$ have significantly less influence to the total attention flow than $b$.

## 4.2 Head Task Analysis

**LTL trace prediction.** We experimented with predicting satisfying traces to linear-time temporal logic (LTL) (Pnueli, 1977). We used a Transformer trained on this task by Hahn et al. (2021). LTL generalizes propositional logic with temporal operators such as $\bigcirc$ (next) or $\mathcal{U}$ (until) and is used to specify the behavior of systems that interact with their environments over time. An LTL formula is satisfied by a trace, which is an infinite sequence of propositions that hold at discrete timesteps. We finitely represent satisfying traces to LTL formulas as a prefix, followed by a loop, denoted by curly brackets. For example, the LTL formula $\bigcirc(a \wedge \bigcirc \neg a)$ denotes that in the second position, $a$ must be true, and in the third position $a$ must be false. The model correctly predicts the trace, where the first position and the loop are arbitrary and hence set to true: $trace : 1; a; \neg a; \{1\}$. Figure 5b depicts the maxflow computation for two heads. The left head focuses on the $\bigcirc \neg a$ part of the formula and

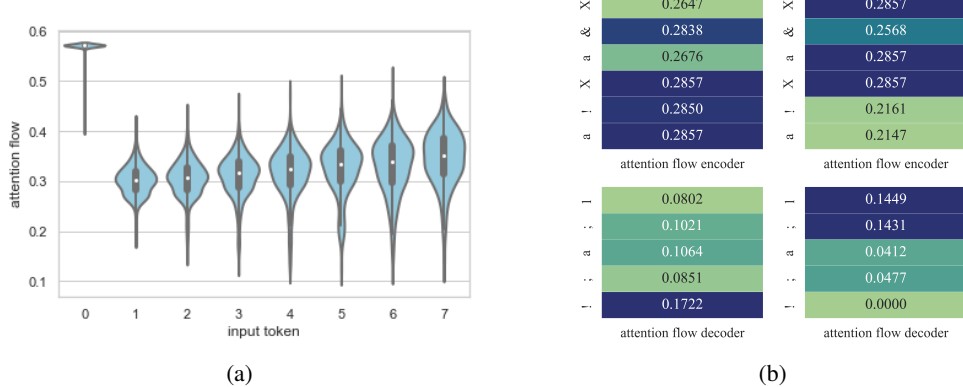

(a)                    (b)

Figure 5: Violinplot for the distribution of attention flow of GPT-2 for 500 samples in (a) and Heatmaps for two heads of the LTLSat model, each attending a different timestep in (b).

the third position of the trace where $a$ is *not* allowed. The right head focuses on the left conjunct $a$, which must appear on the second position of the trace (see Appendix B for another example).

**Translation.** In this experiment, we used the OPUS-MT-EN-DE model Tiedemann and Thottingal (2020) for translating between English and German. The input sentence is "The pilot lost her suitcase.", which is translated to "Der Pilot hat ihren Koffer verloren". The computed flow network can be found in Figure 12. While the meaning of the original sentence is ambiguous, as "the pilot" could be male or female, the translated sentence is not, since the German phrase *Der Pilot* means a male pilot. It has been conjectured that such gender-biased translations can facilitate problematic stereotypes (Bolukbasi et al., 2016). Our analysis technique allows further insight into the internal mechanics of the Transformer model in such a scenario. We analyze the task of the heads, two of them are shown in Figure 6. By computing the attention flow for the encoder and decoder, we can observe that the depicted heads solve opposing tasks: The head on the left-hand side attends *pilot lost her* in the encoder and *Der Pilot* in the decoder, which is the one-to-one translation, but without a corresponding possessive pronoun. The head on the right-hand side attends *pilot* and *suitcase* in the encoder and *Pilot hat* as well as *Koffer* in the decoder. Hence, from the attention flow, we can see that the second head has little influence on the biased translation, as neither *her*, nor *Der* and *ihren* (the German pronoun corresponding to *her*) receive significant attention. This approach, therefore, gives us a helpful hint that we have to analyze the first head to get to the root of this biased translation.

**Head attention.** We analyze the influence of each head of GPT2 based on their contribution to the attention flow. Figure 4a shows the attention flow for each token and head for the input and output sentence *"My name is John, my profession is to be a doctor. I am a doctor of medicine."*. Heads 0, 1, and 2 show high and diverse attention flow values for different tokens, whereas all other heads have shallow and stable attention flow values. To explore this further, Figure 4b shows the accumulated attention flows for all tokens and each head for 300 random samples. It supports the claim that the first three heads have higher attention flow values than all other heads.

| Input | Negative | Neutral | Positive |
|---|---|---|---|
| John is a killer. | 0.9548 | 0.0417 | 0.0034 |
| John is a good killer. | 0.8949 | 0.0967 | 0.0084 |
| John is a good killer 😊 | 0.0981 | 0.3166 | 0.5853 |

(a)

| Network | Heads | Layers | Architecture |
|---|---|---|---|
| DialogPT-medium (MIT) | 16 | 24 | *dec.* |
| OPUS-MT-EN-DE (MIT) | 6 | 8, 8 | *enc. + dec.* |
| PropSat (MIT) | 4 | 4, 4 | *enc. + dec.* |
| LTLSat (MIT) | 4 | 4,4 | *enc.+ dec.* |
| GPT-2 (MIT) | 12 | 12 | *dec.* |
| RoBERTa (MIT) | 12 | 12 | *enc.* |

(b)

Table 1: Results of the sentiment analysis in (a) and the parameter overview of the models in (b).

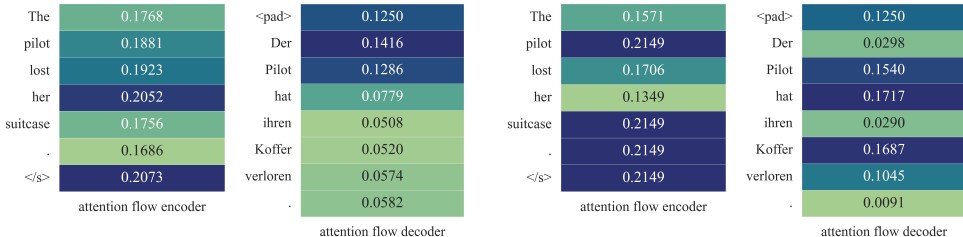

Figure 6: Heatmap for two heads, divided in encoder and decoder. The left head attends the pilot, the head on the right the suitcase.

### 4.3 BIAS DETECTION

**Sentiment detection.** The flow analysis can be used to detect biases in the Transformer models. In this experiment, we used RoBERTa Liu et al. (2019) finetuned for sentiment analysis on the TweetEval Barbieri et al. (2020) benchmark and computed the influence of input tokens on the total flow deciding the classification. We computed the attention flow values for the input tokens of the following sentences and their results, shown in Tab. 1a The resulting flow network can be found in Figure 11 in the appendix. While the first two sentences, "John is a killer." and "John is a good killer." are correctly labeled with negative sentiment (even when having the adjective "good" in the sentence), having an emoji in the sentence immediately shifts the sentiment to be (falsely) labeled as positive. The computation of the attention flow is visualized in Figure 10 in the appendix. For the first two sentences, the attention on *killer* is the highest, considering only non-special tokens. Although the same holds for the third sentence, i.e., the attention flow denotes killer as the most important word, the low-attended smiley changes the sentiment to *positive*. When computing the attention flow for each head, we observe heads with an attention flow of 1.0 to the emoji (see Figure 9 in the appendix).

**First token bias.** While analyzing the attention flow of the decoder-only Transformer DialogPT Zhang et al. (2020) and GPT-2 Radford et al. (2019), we observed a heavy bias toward the first decoded token (see Figure 3a and Figure 9 in the appendix). We computed the attention flow for 500 random samples of the `OPUS-MT-EN-DE` test set. The results are visualized in Figure 5a. The first token contributes the most to the total attention flow regardless of the input tokens. Since the DialogPT model was trained on a dataset mined from `reddit.com`, it might be beneficial to overattend the first token as many conversations on `reddit.com` consist of concise sentences or even single words. One should be aware of this bias when applying this model outside of similar domains.

## 5 LIMITATIONS AND CONCLUSION

The main limiting factor of this approach is that the attention flow in a Transformer is the largest but not the only factor for deciding the next token prediction. Additionally to the many residual connections (which can be incorporated into the flow networks; see Section 2), Transformer models contain feed-forward networks used as intermediate steps. Another minor caveat is that flow values cannot be compared across different model architectures as their absolute values have no meaning. The values can solely be compared to other tokens in the same layer of the same model. This approach should thus be seen as a valuable addition (not a replacement) to the large toolbox for interpreting machine learning models. It generalizes the efforts in visualizing and interpreting raw attention values and attenion rollout. During our experiments, we found the attention flow values computed with the presented approach instrumental in analyzing models, finding biases, and fixing respective datasets.

To conclude, we formalized and extended the technique to construct a flow network from the attention values of encoder-only Transformer models to general Transformer models, including an auto-regressive decoder. Running a maxflow algorithm on these constructions returns Shapley values that determine the impact of a token on the total attention flow leading to the decoder's decision. We provide an implementation of our approach that can be applied to arbitrary Transformer models. Our experiments show this analysis method's applicability in various application domains. We hope our implementation and constructions presented in this paper will aid machine learning practitioners and researchers in designing reliable and interpretable Transformer models.

## 6 REPRODUCIBILITY STATEMENT

The supplementary material of this submission includes python notebooks to reproduce the figures presented in this paper with their underlying data. The code, datasets, models, and our notebooks for the reproduction of the experiments will be made publically available once the double-blind reviewing process ends.

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

## A    SHAPLEY VALUE PROPERTIES

1. **Efficiency.** All of the available payoff $v(P)$ is distributed between the players: $v(P) = \sum_{i \in P} \varphi_i(P, v)$.

2. **Symmetry.** Two players that have the same impact on the total payoff when joining a coalition, receive the same share of the payoff: $\forall S \subseteq P \backslash \{i, j\}.\ v(S \cup \{i\}) = v(S \cup \{j\}) \rightarrow \varphi_i(P, v) = \varphi_j(P, v)$.

3. **Null Player.** A player that has zero impact upon joining a coalition, receives no share of the total payoff: $\forall S \subseteq P \backslash \{i\}.\ v(S) = v(S \cup \{i\}) \rightarrow \varphi_i(P, v) = 0$.

4. **Additivity.** The share of a player in TU game $(P, v + w)$ is the sum of their shares in games $(P, v)$ and $(P, w)$: $\forall i \in P.\ \varphi_i(P, v + w) = \varphi_i(P, v) + \varphi_i(P, w)$.

## B    HEAD TASK ANALYSIS: LTL UNTIL-OPERATOR

In this experiment, we provide another LTL example where one of the heads is focusing on the temporal operator in the formula and another is focusing solely on the propositions of the formula (see Figure 7) The input formula is: $a \mathcal{U} b \wedge 1 \mathcal{U} a$, where $1 \mathcal{U} a$ denotes that finally an $a$ must occur. The network correctly outputs the following trace: $trace : a \wedge b; \{1\}$.

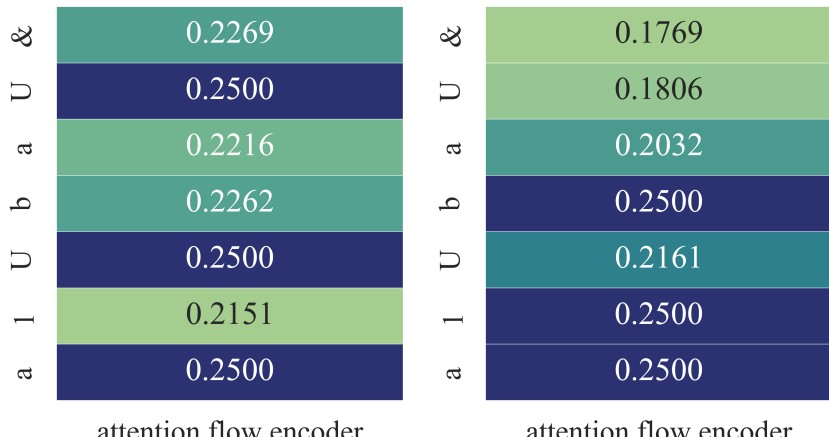

Figure 7: Heatmap of the attention flow for two heads: The left head focuses on the until-operator and the right head focuses on the propositions.

## C    ADDITIONAL FIGURES

### C.1    HEATMAPS

**Single Head Attention Flow in RoBERTa.**    Figure 8 depicts the attention flow of the first head in the RoBERTa model. Intuitively, the word *killer* dominates the sentiment of the sentence. However, the output of RoBERTa is a positive sentiment, although the attention flow is mainly on the word *killer* (see Figure 10). Analyzing the individual heads, one can observe that head 0 attends the smiley with its maximal value (1.0), which could be one explanation for the output of the model.

**Bias in DialogPT.**    Figure 9 shows the attention flow from each token to the current output. While we observe slight changes of the computed attention flow for each token, the first input token *The* is highly attended, more than two times the attention flow than any other token. Note that this observation does not directly translate into a bias in the model, it solely shows that the distribution of attention is biased.

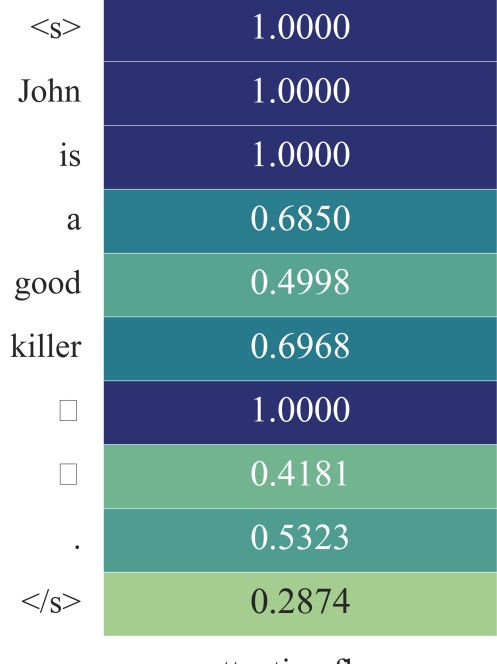

Figure 8: Heatmap showing head 0 of RoBERTa for the example in Figure 10.

## C.2 FLOW NETWORKS

**Encoder Only.** Figure 11 shows the flow network for RoBERTa. The underlying architecture is encoder only with 12 layers, represented by the 12 layers of attention edges between the nodes, and an input sentence with 10 tokens, represented on the y-axis of the network. The special property of RoBERTa is the classification token at position 0 - only the attention flow to this token in the last node layer is important.

**Encoder Decoder.** Figure 12 shows the flow network for OPUS-MT-EN-DE. The underlying architecture consists of an encoder and a decoder with 8 layers each, connected by the cross attention edges in between. For each input token and auto regressive token, we compute the attention flow to each predicted token. In Figure 12, the attention flow for the third predicted token is computed.

**Decoder Only.** Figure 13 shows the flow network for GPT-2 with the underlying decoder only architecture. The model has 12 layers, attention can only flow from previous auto regressive tokens, including the input tokens. We start computing attention flow for the first output token, which is connected to the terminal node in Figure 13.

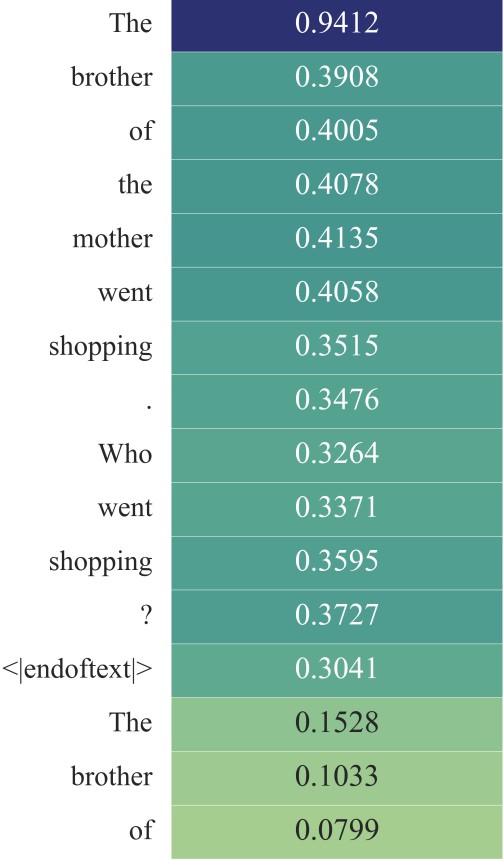

Figure 9: Heatmap showing the bias towards the first token in DialogPT.

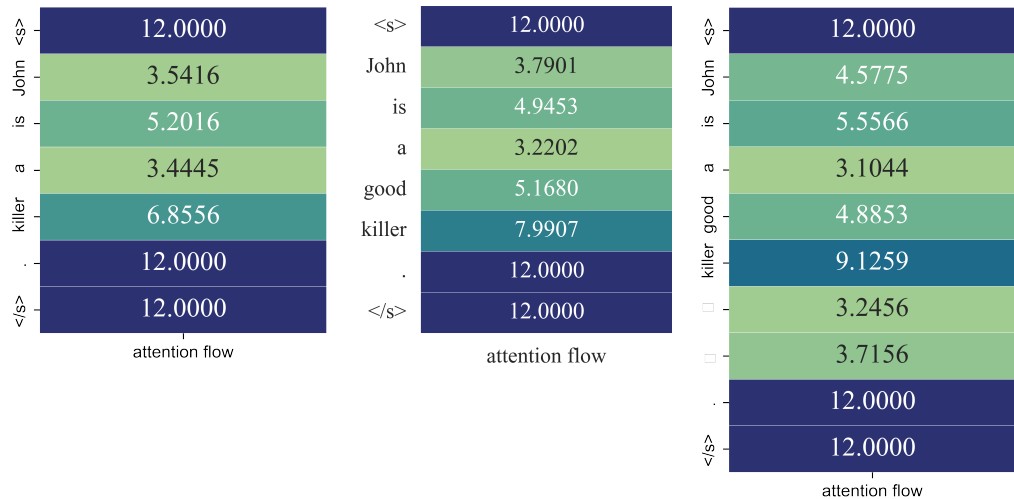

Figure 10: Heatmap showing the attention flow for 3 variations of the same sentence in RoBERTa.

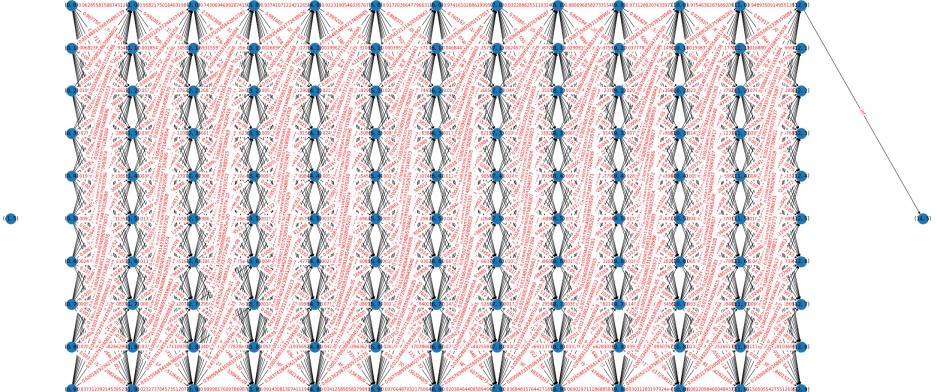

Figure 11: The flow network of the encoder-only network RoBERTa for the example in Figure 10.

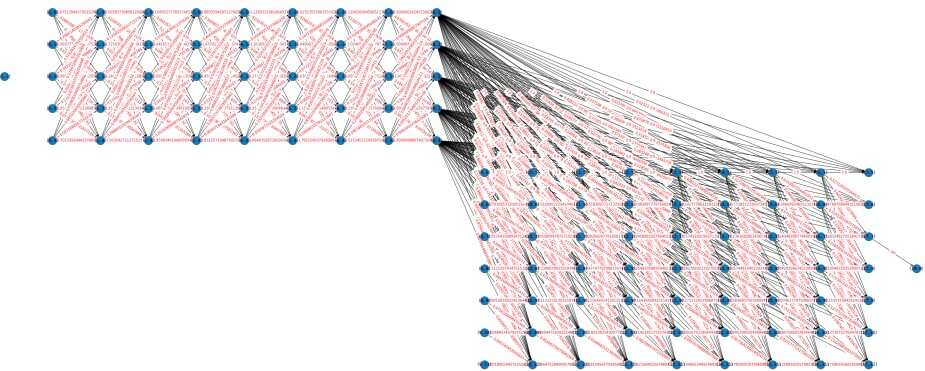

Figure 12: The flow network of the encoder decoder architecture OPUS-MT-EN-DE for the input "The father cooked dinner." and the predicted tokens "Der Vater kochte Abendessen".

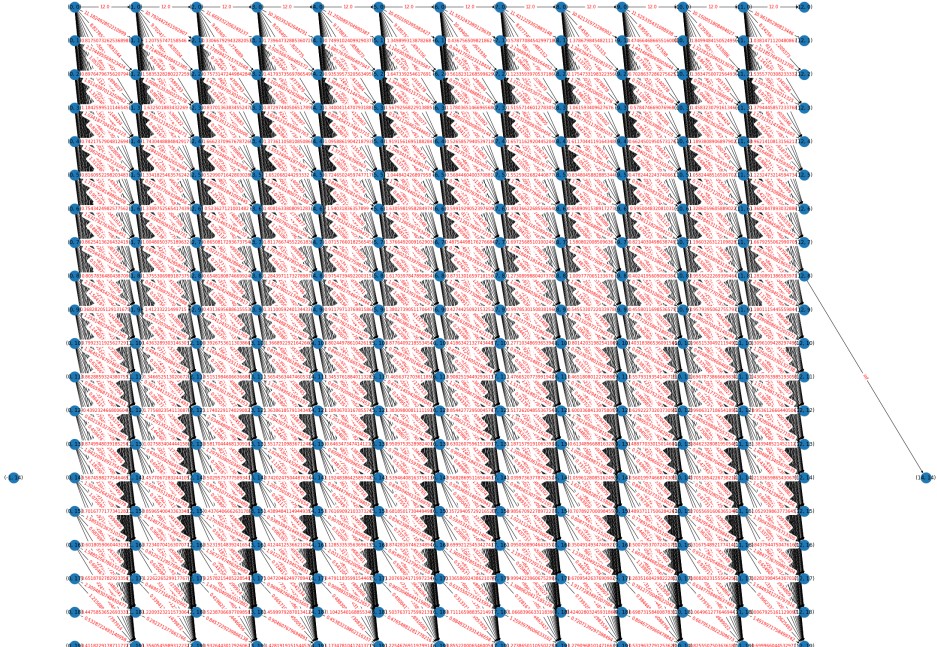

Figure 13: The flow network for the decoder only architecture GPT-2 for the example in Figure 3a.

