# OpenReview forum: "Attention Flows for General Transformers"
_ICLR.cc/2023/Conference — Submitted to ICLR 2023_

### Official Review · Reviewer_uJJx · 2022-10-20

**Confidence:** 4
**Correctness:** 4
**Technical Novelty And Significance:** 2
**Empirical Novelty And Significance:** 2
**Recommendation:** 3

**Clarity, Quality, Novelty And Reproducibility:**

The paper makes an effort to formalize the problem and make connections with game theoretical concepts. I'm not sure these connections are worth the consequent over-technification of the paper, even though it is claimed to be one of the central contributions. Maybe more examples or experiments showing how these scores behave in practice or could be feed as inputs to something else could be more useful.

In general, the paper reads well and smoothly.

The authors will open source the code eventually, and thus results should be reproducible.

In terms of novelty, the paper ends the introduction with a related-work paragraph. This work mainly builds on top of two previous works [1, 2]. [1] introduces attention flows (mainly for Transformer encoders), and [2] shows attention flows are Shapley values for a specific formulation of a cooperative game. My understanding is that the contribution of this paper is to extend the ideas in [1, 2] to decoder-only and encoder-decoder models, for example by handling temporal dependence in autoregressive models (e.g. positional independence).

Providing the code will be useful for the ML community, to explore and analyze trained models.

[1] = Abnar and Zuidema, Quantifying Attention Flow in Transformers.
[2] = Ethayarajh and Jurafsky, Attention Flows are Shapley Value Explanations.

**Strength And Weaknesses:**

The overall approach makes intuitive sense (note it isn't new), but the outputs are numbers that are still hard to interpret or use, as there are many moving parts. After going over the paper, I ended up with the impression of having read about a technical tool that isn't solving any specific problem. Given the amount of follow-up work on [1], it seems the ML community probably has found applications for it, though.

I appreciate the authors' efforts and honesty at the end of the paper to list limitations of the work. One example is the fact that Transformer models also apply layers other than attention, like MLPs. I'm not sure if the proposed method accounts for token transformations via those; maybe the fact that they affect the subsequent attention scores is enough? i.e. if an MLP layer zeroes-out a token, the attention scores used to solve the max-flow problem (matrix A in the paper) will propagate the zeros, right? Otherwise results would be misleading.

In terms of experiments, as a proposal, I think a simple vision example should be much more informative. Each input token in VIT corresponds to a patch, and for image classification, the authors could show the input image superimposed with the corresponding attention score of each patch. Ideally, the attention centers around the object that is to be classified. It would also be interesting to see how solving a max-flow problem with hundreds of input tokens (like in VIT) scales in terms of runtime.

The authors offer some time estimates in 4.1 (text completion), but given the complexity and engineering required to compute attention flows, I was expecting a more in-depth analysis of how expensive it is to compute these numbers for real-world models. Grouping inputs or skipping layers, as mentioned in Section 2.5 is an interesting approach, but no evidence is provided on that direction suggesting it could work.

It's hard to conclude much from some of the experiments (like Figure 4 or 5a).

[1] = Abnar and Zuidema, Quantifying Attention Flow in Transformers.

**Summary Of The Paper:**

The paper proposes a method to quantify the impact of each input token on the outcome of a Transformer model. The algorithm constructs a flow network whose edges contain attention values, and solves the max flow problem for each vertex corresponding to an input token. The idea is applied to encoder only, decoder only, and encoder-decoder models.

The paper makes some connections with game theoretical concepts to justify the meaningfulness of the approach and the computed scores.

Finally, the paper shows some experiments where the token-importance values are computed, and offers an explanation regarding the underlying logic or intuition behind each example.

**Summary Of The Review:**

This paper extends [1, 2] to decoder only and encoder-decoder architectures, and provides some experiments where the method is applied. The contribution of the paper seems modest, and maybe some of the mathematical formalization could be replaced with more examples or applications.

[1] = Abnar and Zuidema, Quantifying Attention Flow in Transformers.
[2] = Ethayarajh and Jurafsky, Attention Flows are Shapley Value Explanations.

---

> ### Comment · Reviewer_uJJx · 2022-11-21
> **Rebuttal**
>
> I would like to thank the authors for the 'General Rebuttal' above, and acknowledge their remark on the paper contributions.
>
> However, I won't update my score at this point.

---

### Official Review · Reviewer_DmXz · 2022-10-22

**Confidence:** 4
**Correctness:** 3
**Technical Novelty And Significance:** 2
**Empirical Novelty And Significance:** 3
**Recommendation:** 3

**Clarity, Quality, Novelty And Reproducibility:**

The paper is clearly written, with clear motivation and overall structure. The code provided is also workable.

**Strength And Weaknesses:**

The strength of this paper lies in the following aspects:

1. Although the attention flow formulation for transformer encoder has been proposed before, this paper extends this to cover the rest of two architectures so that this method can be applied to auto-regressive structure. The model formulation is solid with minor typos.

2. The connections between attention flow values and the Shapley values has been shown before. This work, however, is able to explore the assumption on the positional independence so that the proof is simplified and is able to avoid some issues behind the underlying assumptions of the previous proof.

The weakness of this paper lies in the following aspects:

1. Lack of objective and reliable metric to decide if the attention flow is relevant or not. It is stated in the paper also that the attention flow value is not supposed to be interpreted as causal factors to the prediction or translation. It is merely an associational factor. This together with lack of large scale studies and peer-reviews in terms of the performance of the method limits its application in real life decision making.

2. The proof that the attention flow is Shapley value relies on the effectiveness of the positional independence assumption. A solid verification and analysis is necessary to convince the reader that the positional independence is guaranteed in experiments. It is very challenging in general to check on this assumption even if the weight is normalized for tokens output later.

3. The proof that the attention flow is Shapley value under the positional independence assumption needs to be shown in detail instead of explained vaguely. The contribution of this paper is its mathematical formulation of the attention flow. It is necessary to show this critical proposition. Hope that this part can be added in appendix later on.

**Summary Of The Paper:**

In this paper, the author provides a comprehensive formulation of the attention flows for different architectures of transformers (encoder, decoder, encoder-decoder). In order to account for the auto-regressive decoder structure, the author adjusted the attention flow to ensure the positional independence of the computed maxflow values. Following the previous work, the author further shows that the maximum attention flow corresponds to the Shapley value under the three architectures of transformers.

**Summary Of The Review:**

Explaining the decision from large language models has always been a focus since the popularity of these models. This paper improves on top of similar existing works but providing a comprehensive formulation on the attention flow that covers decoder and encode-decoder too. It is of great interest to this community to see progress that leads to better explainability of the model. On the other hand, this paper's main contribution is not significant enough for ICLR since the major works (both the attention flow and the equivalence to Shapley value) exist before and it did not contribute significantly outside the existing framework. The performance evaluation is also tricky since only a few examples are shown but lacks of more experiments to demonstrate the strength of attention flow method. I would like to see some improvement on the matter.

---

### Official Review · Reviewer_4MwZ · 2022-10-24

**Confidence:** 3
**Correctness:** 2
**Technical Novelty And Significance:** 2
**Empirical Novelty And Significance:** 2
**Recommendation:** 3

**Clarity, Quality, Novelty And Reproducibility:**

I finished the paper unsure of the impact of its contribution. The paper is not clear on the motivation for their method, and I gather no new insights from their analysis on the attention of Transformers.

**Strength And Weaknesses:**

**Strengths**

- (+) Authors explain that Attention Flow can be connected between an encoder and decoder of a Transformer.

**Weaknesses**

- (- -) The paper’s analysis is very limited. <5 sentences are analyzed in the main paper to show that their method works. The insights they uncover from the shapley values about the role that individual heads play in transformer attention are not novel.
- (- -) The visualization library seems limited and not thoroughly introduced even though it was a central part of the abstract. The paper’s figures simply show tokens next to shapley values as colored blocks. The background color of the tokens overemphasizes small differences in the shapley value when all values are similar and is more deceiving than helpful. Additionally, the fact that attention flow primarily attends to punctuation and special tokens is not meaningful to understanding the data domain.
- (-) Across the board, I am unable to understand why the experimental results are particularly meaningful. The experimental section feels more like showing that you can get numbers at all using this method.

 **Other Comments**

- Statement “In practice, however, heads are biased towards keeping their respective tasks” [across layers] is not defended, and it has been my experience that in Transformers this is entirely not the case. In Transformers that have no weight sharing, there is no mechanism that encourages a head to perform the same association function across layers.
- Figs 4 and 5 are not self contained. I cannot see the input tokens in the plot itself, making the results difficult to interpret.
- Figure 2’s explanation on the top of page 4, “encoder (top) and a decoder (bottom)” is inconsistent with the design of the figure itself.
- Unclear what “positional independence” (sec 1 page 2) of computed maxflow values means — positional encoding is embedded into the token representation itself and attention learns to attend to each token+position? I could not find where this was further explained in the paper

**Summary Of The Paper:**

The paper constructs an attention flow network out of encoder (& decoder) Transformers, and release a library to visualize the attention flow of Transformers.

**Summary Of The Review:**

Overall, the paper feels like an extension to the original Attention Flows paper and not like a standalone paper. The experiments are limited, and the knowledge uncovered by the visualizations and methods are not impactful.

I did not read the Appendices thoroughly.

---

> ### Comment · Reviewer_4MwZ · 2022-11-21
> **Re: General Rebuttal**
>
> I thank the authors for their general rebuttal. However, the clarifications do not address my primary objections with the paper itself. My score remains unchanged.

---

### Official Review · Reviewer_wsJr · 2022-10-27

**Confidence:** 3
**Correctness:** 3
**Technical Novelty And Significance:** 2
**Empirical Novelty And Significance:** 2
**Recommendation:** 5

**Clarity, Quality, Novelty And Reproducibility:**

This paper is original and well-written. However, I think the connection to Shapley values in its current form is over-complicating things and it's not necessary for the understanding of the paper.

**Strength And Weaknesses:**

Strengths:
1. Extends attention flow to encoder-decoder and decoder-only transformers.
2. Empirical analysis of token importance on several tasks.

Weaknesses:
1. The change to Abnar and Zuidema 2020 seems incremental.
2. The connection to Shapley values is drawn by defining value function to be based on maxflows, so the equivalence is not surprising at all. A more interesting connection would be to use the actual log probability of the target token as payoff values and see if there's a correlation.
3. Related to 2, it is not clear if attention maxflows reflect actual token importance. It would be more convincing if the maxflow values can be compared against feature importance obtained using other methods (such as Shapley values using log probability of target token as payoffs, or simply gradient-based saliency maps). For example, does the first-token bias found by maxflow hold for Shapley values?
4. The analysis on individual attention heads doesn't make that much sense to me, since the same head id across different layers does not mean they have anything in common.

**Summary Of The Paper:**

This work extends the attention flow method proposed in Abnar and Zuidema 2020 to encoder-decoder and decoder-only transformers. The major contribution is based on the observation that later predicted words have more incoming edges than earlier words, such that to ensure positional independence, this work proposes a method to normalize maxflow values. In addition, this work draws connection between maxflow attention and Shapley values by defining payoffs as the sum of maxflows and showing the equivalence under this definition. Experiments on several tasks show that this method is able to gain insights into token importance for a prediction task.

**Summary Of The Review:**

My major concerns are: 1. the change to the original attention flow paper is incremental; and 2. a correlation study with feature importance found by other methods (such as Shapley values) is missing and it's not clear if the found maxflow values mean anything. Therefore, I'm leaning towards rejecting this paper.

---

> ### Comment · Reviewer_wsJr · 2022-11-21
> **Post Rebuttal**
>
> I would like to thank the authors for their rebuttal. While I do understand this paper extends Abnar and Zuidema 2020 to a decoder setting, and that it connects to Shapley values in a way that no prior works did, I still think these extensions are quite straightforward, and that connecting to Shapley values by defining the value function based on maxflows is not surprising and not useful. Therefore, I'm keeping my score unchanged.

---

### Author Response · Authors · 2022-11-18
**General Rebuttal**

We appreciate the time that the reviewers have taken to review our submission and thank them for their suggestions for improving the paper.
We want to point out several points on which we disagree with the assessment:

1) The approach “only” extends previous work:

This is only one of the paper’s contributions. The contributions of this paper are:
- Formalizing the attention-flow constructions mathematically for easier extendability, implementation, and to assess the validity of the claims made in previous work.
- Provide an extension to Transformer decoders.
- Re-drawing the connection to Shapley values. Note that our line of reasoning significantly differs from Ethayarajh and Jurafsky (2021) (page 6), based on uncovering shortfalls in their reasoning after formalizing the approach rigorously.
- Provide an easy-to-use implementation that can be incorporated in larger explanation frameworks, e.g., by replacing vanilla attention value visualizations, which are still being used today.

2) Limited experimental results:

The experimental evaluation section is structured as a proof of concept in different domains. Interpretability of the resulting values heavily depends on the domain. For example, applying the technique to SAT requires different interpretability than application on natural language.
The approach, thus, must be interpreted in a larger framework (see limitations). We do not consider our contribution less valuable, as this also holds for any previous work on this topic. Providing fully-fledged explainability tools for every domain is beyond the scope of this work.

---

### Decision · Program_Chairs · 2023-01-20

**Decision:**

Reject

**Justification For Why Not Higher Score:**

limited contribution, see meta review for details.

**Justification For Why Not Lower Score:**

Reject

**Metareview: Summary, Strengths And Weaknesses:**

This paper studies the computation of how much an input token in a Transformer model influences its prediction. The authors formalize a method to construct a flow network out of the attention values of encoder-only Transformer models and extend it to general Transformer architectures.

Strength
- Extends attention flow to encoder-decoder and decoder-only transformers.
- Empirical analysis of token importance on several tasks.

Weakness
- The extension from Abnar and Zuidema 2020 seems incremental.
- The visualization library seems limited and not thoroughly introduced even though it was a central part of the abstract.

Please consider the comments received to improve the next iterations of the paper.